# Comparative Analysis of Machine Learning Methods for Predicting Robotized Incremental Metal Sheet Forming Force

**DOI:** 10.3390/s22010018

**Published:** 2021-12-21

**Authors:** Vytautas Ostasevicius, Ieva Paleviciute, Agne Paulauskaite-Taraseviciene, Vytautas Jurenas, Darius Eidukynas, Laura Kizauskiene

**Affiliations:** 1Institute of Mechatronics, Kaunas University of Technology, 51424 Kaunas, Lithuania; ieva.paleviciute@ktu.edu (I.P.); vytautas.jurenas@ktu.lt (V.J.); darius.eidukynas@ktu.lt (D.E.); 2Department of Applied Informatics, Kaunas University of Technology, 51368 Kaunas, Lithuania; agne.paulauskaite-taraseviciene@ktu.lt; 3Department of Computer Sciences, Kaunas University of Technology, 51368 Kaunas, Lithuania; laura.kizauskiene@ktu.lt

**Keywords:** incremental sheet forming, failure prevention, friction force, robotized manufacturing, prediction model

## Abstract

This paper proposes a method for extracting information from the parameters of a single point incremental forming (SPIF) process. The measurement of the forming force using this technology helps to avoid failures, identify optimal processes, and to implement routine control. Since forming forces are also dependent on the friction between the tool and the sheet metal, an innovative solution has been proposed to actively control the friction forces by modulating the vibrations that replace the environmentally unfriendly lubrication of contact surfaces. This study focuses on the influence of mechanical properties, process parameters and sheet thickness on the maximum forming force. Artificial Neural Network (ANN) and different machine learning (ML) algorithms have been applied to develop an efficient force prediction model. The predicted forces agreed reasonably well with the experimental results. Assuming that the variability of each input function is characterized by a normal distribution, sampling data were generated. The applicability of the models in an industrial environment is due to their relatively high performance and the ability to balance model bias and variance. The results indicate that ANN and Gaussian process regression (GPR) have been identified as the most efficient methods for developing forming force prediction models.

## 1. Introduction

The forces exerted by the friction between the tool and the workpiece during the machining of materials play an important role in the quality of the product. The effect of petroleum and vegetable oil-based lubricants on the friction coefficient, wear, forming forces, and surface roughness of metal sheets produced by the single point incremental forming (SPIF) process has been investigated in [1]. Lubricating oil was found to produce the surface roughness in the direction perpendicular to which the tool passes, with a value close to Ra = 1.45 μm, which exceeds the surface roughness of the undeformed sheet while degrading the quality of the final product. As the lubricants are unfriendly to the environment, new methods for reducing and predicting forming forces have to be found. With the proliferation of data in recent technologies, machine learning (ML) has become one of the most important methodological approaches for extracting meaningful insights from huge amounts of data. The authors in [2] present an ML algorithm-based method for predicting the occurrence of defects in the SPIF process of metal sheets due to material properties and sources of dispersion of process parameters. The defects and failures are directly related to the tensile flow behaviour of the material sheet, which is predicted using phenomenological models and Artificial Neural Networks (ANN) [3]. Using a genetic algorithm and linear regression analysis, the constants of the Johnson-Cook, Khan-Huang-Liang, and modified Voce equations were calculated and used to model uniaxial tension tests. The modified Voce constitutional equation was found to predict the flow behaviour of AA5182-O better than other models. ANN models can be trained and developed for the final evaluation of processes and tools prior to production [4].

The SPIF process aims to form products with the most accurate shape possible [5,6,7,8,9]. A deep learning technique to propagate geometric accuracy in SPIF was proposed in [5]. The prediction of geometric accuracy is one of the most important indicators of product quality. In order to predict geometric accuracy in SPIF, shallow learning and deep learning methods are investigated and compared. In [6], a Modified Adaptive Neuro-Fuzzy Inference System (MANFIS) focuses on geometric deviation prediction, and an Enhanced Squirrel Search Algorithm (ESSA) is used for the optimal selection of SPIF parameters in AA2024-O aluminum alloy sheets. Several shaping experiments have been carried out using SPIF under different forming conditions to measure the surface roughness, the arithmetic mean roughness (Ra), and ten-point mean roughness (Rz) of the AlMn1Mg1 sheet [7]. In addition, an ANN was used to predict (Ra) and (Rz), given the data collected from the SPIF components. Extensive experimental work was carried out and presented in [8] to investigate compliance with industry requirements. The influence of forming parameters (forming wall angle, step depth, and feed rate) was investigated to achieve SPIF capabilities at higher speeds for forming aluminum alloy AA5754-H22 and DC04 steels. Soft computing-based simulation and multi-functional optimization of the process parameters using an aluminum alloy sheet SPIF to obtain the desired deformed shape, optimally formed to meet several objectives is presented [9]. A response surface methodology and Adaptive Neuro-Fuzzy Inference System (ANFIS) models were developed to predict the responses based on experimental data collected from a central composite experimental design, taking into account feed rate, tool diameter, and step height as inputs and outputs, i.e., deformed sheet thickness, forming wall angle, and surface roughness.

One of the indicators of the quality of the formed product is the surface quality [10,11,12,13,14,15]. The authors in [10] present the results of an analysis of the interaction between the SPIF process parameters and the stiffened ribs made of Alclad aluminum alloy plates. The correlation between the operating variables, the maximum forming angle, and the surface roughness is determined by ensemble learning using a gradient boosting regression tree and presented in [11]. To obtain the ML dataset, a series of experiments were conducted with a continuous variable-angle pyramid shape based on D-Optimal design. In [12], the authors describe an experimental study on the influence of process parameters on the surface roughness of SPIF using a dummy sheet. The authors in [13] propose the use of Support Vector Machine (SVM) algorithms to generate models that can predict the geometric accuracy of SPIF molds made from DX51 aluminized steel sheets. The data obtained using a coordinate measuring machine were used to train the generated models. In the study [14], a multiple-layer perceptron-type ANN model is used to predict the SPIF quality part and reduce the gap between the shape obtained using a CNC machine and the CAD. In [15], different ANN models and structures were presented to predict the accuracy and formability of SPIF components made from thin aluminum alloy blanks. The main finding of the study revealed that the structure of a single output solution showed better results than a network with two outputs. Within the scope of the experiments, the results were evaluated using different validation metrics: the highest R2 values were 0.9909 and 0.9860, and the lowest MSE values were 0.1503 and 0.0351 for accuracy and formability, respectively.

One of the most important parameters of the technological processes is the prediction of the acting force, as it is related to potential defects and surface quality [16,17,18,19,20,21]. In [16], a transfer network is developed using simulation data as the source domain and experimental data as the target domain, in combination with related learning transfer methods and theories. The aim of the study [17] is to predict and optimize the surface roughness and cutting forces in the milling of aluminum alloy 7075-T6 using regression analysis, Support Vector Regression (SVR), ANN, and a multi-objective genetic algorithm. The aim of the research presented in [18] is to investigate the capabilities of SVR, polynomial regression, and ANN to optimize turning parameters. Measuring the forming force using SPIF technology helps to identify optimal processes, avoid failures, and implement online control. The experimental study presented in [19] focuses on the SPIF process and the influence of four different process parameters, i.e., step size, tool diameter, sheet thickness, and feed rate, on the maximum forming force. An ANN and a regression method were used for an efficient force prediction model based on the ANFIS. To develop an efficient data-driven force prediction using back-propagation neural networks, a virtual data generation method based on a mega-trend diffusion function and a particle swarm optimization algorithm was proposed in [20]. The aim was to improve the accuracy of SPIF force prediction based on small experimental data. In [21], an ANN was applied to predict the minimum force required for SPIF sheets of thin aluminum AA3003-0 and calamine brass Cu67Zn33 alloys.

A review of the research papers reveals that until now, there has been no attempt to reduce friction between the tool and the sheet metal by eliminating environmentally hazardous petroleum products, which also affect the chemical composition of the sheet material being formed. Another important issue is that after the forming process, the lubricated surfaces need to be cleaned, which also increases the development time and costs of the product.

This paper proposes an innovative SPIF, by exciting the sheet metal with ultrasonic vibrations, thereby eliminating environmentally hazardous contact surface lubrication. This has led to a reduction in forming forces and the prediction of these forces by different ML algorithms has allowed the selection of the most efficient ML methods capable of performing intelligent data analysis. The main contribution of this study is to explain the principles of different machine learning techniques and their applicability to SPIF. Overall, this paper aims to be a point of reference for academic researchers, industry professionals, as well as decision-makers in the field of production, especially from a technical point of view.

## 2. Materials and Methods

Metallic materials used in the forming process, such as SPIF, visibly deform in contact with the tool area. The key challenge of the SPIF process is to effectively assess the forming forces. The forces generated during SPIF may be controlled by varying the coefficient of friction between the tool and the workpiece. Lubrication is an important factor in the SPIF process, reducing friction in the tool-sheet metal contact area, but the use of grease is associated with environmental problems. Therefore, it is necessary to find other ways to reduce the forming forces associated with the process dynamics. This requires, in particular, the development of a SPIF process for which the mechanical parameters can be adjusted experimentally using the 3D scanning device shown in Figure 1.

### 2.1. Experimental Investigation of Aluminum Sheet Dynamics

During the SPIF process, the tool moves by deforming the aluminum alloy sheet, so the amplitudes and frequencies of the sheet’s eigenmodes are influenced by the position of the tool and the forming depth. Experimental studies have been carried out to evaluate the influence of the forming tool contact locations (points 1 and 2 in Figure 2) on the dynamics of the ultrasonically activated metal sheet.

An aluminum alloy AW1050 (mechanical properties and chemical composition of aluminum alloy presented in Table 1) sheet with dimensions of 350 × 350 × 0.5 mm is rigidly attached to the frame and one side is facing three laser scanning heads (Figure 1). On the other side of the metal sheet, a forming tool with a steel sphere of 8.5 mm radius is in contact with a sheet at the locations presented in Figure 2. Some vibrations of an aluminum alloy sheet are excited in the frequency range from 0.5 to 60 kHz with the piezoceramic transducers attached to a frame. Thereby the 3D frequency response and deformations of an aluminum alloy sheet are obtained.

The presented vibrograms (Figure 3) show that regardless of the location of the tool in contact with the metal sheet and the forming depth, the amplitudes of the Z eigenmodes perpendicular to the sheet surface are significant, while the influence of the tool location on the amplitudes of the X and Y natural oscillation modes in the 2D plane is insignificant.

The experimental results have shown that the SPIF process is most effective in the frequency range from 28 to 36 kHz, where the lateral vibrations in the XY-direction of the sheet dominate over the normal Z-direction vibrations of the sheet. The effect of ultrasonic vibration on the sliding friction of aluminum alloy specimens in sliding tool steel has been studied in [22]. A significant reduction in sliding friction (up to >80%) was observed and good agreement was found between the measured values and the predictions of two simple models for the effects of longitudinal and transverse vibrations. Ultrasound not only reduces friction between the tool and the workpiece but in our developed technology, ultrasound also makes it easier for the tool to slide on the sheet surface. Ultrasonic assisted friction reduction is well known in the field of metal-to-metal contacts. Due to the vibration, the stick phase in the contact phase vanishes and only sliding occurs. As long as the macroscopic relative velocity of the contact partners is much lower than the vibration velocity, the force required to move the parts tends to (nearly) zero. This finding further reaffirms that ultrasonic excitation of the sheet in the 2D plane can have a significant impact on the efficiency of the SPIF process. 

### 2.2. Experimental Investigation of Robotized Incremental Aluminum Sheet Forming

In order to control the forming force effectively, the aluminum alloy sheet was excited by ultrasonic vibrations. An experimental set-up developed for this purpose is shown in Figure 4. The test sheet (1) is clamped to a frame (2) and excited by two piezoelectric transducers (3), attached rigidly to the two opposite sides of the frame. A power amplifier (4) is used to generate the vibrations in the range from 28 to 36 kHz in which lateral vibrations of the sheet are dominant, of the piezoelectric transducer. As the robotic arm (5) moves, the tool with the sphere (6) attached to it incrementally forms the sheet. The tool is specifically mounted at an angle of 30° degrees, so that the mechanical momentum sensor STJ100 (7) is connected to controller BGI (8) (Mark-10 Corp., Copiague, NY, USA) and could measure both—pressing and friction momentum. The sensor controller BGI is connected to an oscilloscope PicoScope 3424 (9) (Pico Technology Ltd., St. Neots, UK, GB), which transmits the results obtained from the mechanical moment sensor on the computer (10).

The parameters of the tool forming path trajectory are given in Table 2.

An illustration of the SPIF product parameters used for the incremental forming of the aluminum sheet is presented in Figure 5.

The mechanical friction between the steel tool and aluminum alloy sheet was measured using torque sensor STJ100, with a sensitivity factor of 6 Nm/V, connected with BGI series digital force gauge (Mark-10 Corp., Copiague, NY, USA) and PC (Figure 4). During this experiment the following set of data was used: speed rate of tool—1200 mm/min, the radius of tool sphere—8.5 mm, normal to sheet surface force—100 N. The averaged measurement results of five tests of the friction coefficient and the friction force of the tool on dry, lubricated, and ultrasonically assisted workpiece surfaces are given in Table 3.

It is clear from Table 3 that under the influence of ultrasonic vibrations the coefficient of friction between the steel tool and the aluminum alloy is close to the coefficient of friction of the lubricated surfaces. This makes it possible to solve environmental problems, and the surface of the manufactured part does not need to be cleaned. In order to assess the effect of vibrations on the SPIF of an aluminum alloy sheet, measurements of the surface roughness of the sheet were conducted with and without ultrasonic vibrations. An advanced surface roughness tester TR200 (BeijingTIME High Technology Ltd., Beijing, China) was used to measure the surface roughness of the metal sheet. The measured surface roughness of the sheet formed without ultrasonic vibrations was in the range Ra = 0.30–0.33 μm and with the ultrasonic vibrations in the range Ra = 0.18–0.25 μm, respectively. This revealed phenomenon has been patented by the authors [23].

## 3. Experimental Data Exploration

The data should be analysed to determine whether the data collected are free of data quality problems that could adversely affect the intended prediction models. Common issues such as missing values and outliers should be calculated because it is impossible to train error-based models with data that contains missing values. Furthermore, data that contain outliers can provide incorrect predictions. As depicted in Table 4, there are two types of features: continuous numeric and categorical binary. Two features, namely “Tool end diameter” and “Wall angle” have constant values, therefore they must be eliminated.

First of all, data exploration should be performed in order to determine whether or not the collected data suffer from any data quality issues that could negatively affect prediction models that are intended to build. No outliers or missing data have been identified in the data set. After the first iteration of data quality issues analysis, the data set contains five inputs—raw features, that come directly from data sources, and one output-vibro excitation of the sheet (Table 5).

The SPIF experiment was performed under dry, lubricated, and vibration-excited friction on the contact surfaces of the tool and the sheet metal, and the vertical force dependences on the process conditions are given in Table 6.

Next, it is reasonable to calculate the correlation coefficients *r*, which indicate the strength of the linear relationship between the two features. The values of Spearman correlation coefficients [24] vary between −1 and 1, whereas if *r* = 0, then the variables have no relationship; the closer the coefficient is to +1 or −1, the stronger the relationship. The sign indicates whether the relationship is positive or negative, e.g., if *r* = 1, then the two features have an ideal positive relationship. A coefficient close to 0 shows a weak correlation. It has been noted that step depth and step width have the same values and a correlation coefficient of *r* = 1. It is thus reasonable to remove one of these two features. No outliers or missing data were identified in our dataset:(1)$$r=\frac{\frac{1}{n}\sum_{i=1}^{n}\left(\left(R\left(x_{i}\right)-\bar{R\left(x\right)}\right)\cdot \bar{R\left(y_{i}\right)-\left(R\left(y\right)\right)}\right)}{\sqrt{\left(\frac{1}{n}\sum_{i=1}^{n}{\left(R\left(x_{i}\right)-R\bar{\left(x_{i}\right)}\right)}^{2}\right)\cdot \left(\frac{1}{n}\sum_{i=1}^{n}{\left(R\left(y_{i}\right)-R\bar{\left(y_{i}\right)}\right)}^{2}\right)}}$$
where R(x) and R(y) are the ranks of the x and y variables; $\bar{R\left(x\right)}$ and $\bar{R\left(y\right)}$ are the mean ranks.

The correlation coefficient between “Tool type” and the output “VFC vibro excitation” is very low because this input attribute is binary, having only two possible values (rotating sphere on the end; not rotating) therefore it is eliminated. After the first iteration of data quality issues management and analysis, our dataset contains six raw features obtained directly from data sources. These correlation coefficients *r* are provided in the Spearman correlation matrix (heatmap) and presented in Figure 6.

It has been observed that “VFC dry friction” has the strongest correlation with the output value, *r* = 0.998. Meanwhile, step depth and sheet thickness, with values of *r* = 0.360 and *r* = 0.433, respectively, have shown a moderate correlation with “VFC vibro excitation”. However, it is reasonable to include all six parameters for the further prediction investigation employing Artificial Neural Networks (ANN).

## 4. Machine Learning Based Prediction

In the field of mechatronics and bioengineering, the size of experimental data sets is often insufficient, thus the prediction task requires machine learning (ML) algorithms capable of generalizing the data properly. Simple models (such as linear regression, decision tree, etc.), feature selection, k-fold for cross-validation [25], ensemble learning, regularization, or possibly, generation of synthetic data [26,27] can be used for this purpose. A number of experimental studies have shown that ANN used for correlation analysis and prediction can yield good results even with a small sample of data [28,29], but other ML algorithms, such as Support Vector Machine (SVM) or Random Forest (RF), are often used as well. In this study, five different supervised machine learning methods were used for the comparative analysis of the prediction results: Gaussian Process Regression (GPR); SVM; Decision Trees (DT); K-Nearest Neighbors algorithm (KNN), and ANN. The obtained results confirmed that ANN is the most accurate method (according to Root Mean Square Error (RMSE)) in the framework of this task (Figure 7a), although it is the least efficient in terms of training time. The k-fold cross-validation procedure shows that the RMSE values of all ML algorithms do not differ significantly. Evaluating the accuracy results, it can be seen that ANN and GPR provide similar performance, but the training time for both algorithms differs greatly. ANN has an average training time of 12.68 s which is significantly higher compared to GPR (Figure 7b). This is because ANN has more parameters than GPR. ML algorithms may give different prediction accuracy and training duration each time, even when trained on the same data set. It is possible to reduce the variance of the ML algorithm by optimizing its hyperparameters.

Hyperparameters, in contrast to ML model parameters, are set manually before the model starts training. Hyperparameters cannot be learned within the estimator directly, however, model parameters are properties of the training data that are estimated automatically. For example, the minimum leaf size in a decision tree, or kernel scale and function of SVM are hyperparameters while the weights in an ANN are model parameters learned during training. The choice of hyperparameters in the above models can strongly affect its performance, therefore the optimization process allows to automatically find the optimal combination of hyperparameters for the ML algorithm [30]. As the result, an optimal model is provided, which reduces a predefined error value and in turn, increases the accuracy of independent data.

### Hyperparameter Optimization

Hyperparameter optimization has been performed on GPR, SVM, DT, KNN, and ANN models using Bayesian optimization [31]. Two other popular hyperparameter tuning algorithms are grid search and random search. Grid search is the simplest algorithm for hyperparameter tuning, which divides the domain of the hyperparameters into a discrete grid. Theoretically, this algorithm should find the best point in the domain, but practically is not used very often, because it is an exhaustive and time-consuming search. Random search, unlike grid search, does not search solution for every possible combination of hyperparameter values but tests only a randomly selected subset of these values. Instead of random searching in the hyperparameter domain, Bayesian optimization enables an intelligent manner of hyperparameters selection, because it uses the results from the previous iteration to decide what is the next set of hyperparameters, which will improve the model performance. Prioritizing hyperparameters is very efficient and allows for finding the best values of hyperparameters’ sets much faster compared to both grid search and random search.

The Bayesian optimization method for the tuning of hyperparameters employs the acquisition function with the purpose to determine the next set of hyperparameter values. There are many different acquisition functions such as upper confidence bound, entropy search, probability of improvement, and expected improvement, but the last two functions are most commonly used. In general, the expected improvement function evaluates the expected amount of improvement in the objective function:(2)$$El\left(x\right)=E[\max\left(0,\;f^{\prime }-f\left(x\right)\right]$$
where $f^{\prime }$ is the minimum value of f observed so far; *x* is the location of that sample.

The performance of such an optimization process depends not only on the chosen acquisition function but also on the surrogate model that helps to approximate the main target functions. In our case, the Gaussian process (GP) has been used, which is the most often preferred choice. In general, the Bayesian optimization follows the sequence of four cycle steps: (1) use Bayes rule to obtain the posterior; (2) choose a surrogate model; (3) use an acquisition function to decide the next sample point; (4) add new data to the set of observations and go to step 2.

Four hyperparameters, i.e., sigma value, basic function, kernel function, and kernel scale, have been included in the optimization process of the GPR model. The kernel function plays a significant role because the choice of kernel functions determines almost all the generalization properties of the GPR model. The sigma value σ is selected within the range calculated by Equation (3).
(3)$$\sigma =\left[\min;\;\max\right]=\left[0.0001;\;\max\left(10\times \sqrt{\overset{n}{\underset{i=1}{\sum }}\frac{\left(y_{i}-\bar{y}\right)}{n-1}}\;\right)\right]$$
where $\bar{y}$ is a sample mean (output sample mean), $y_{i}$—the value from the output sample, *n*—sample size.

The GPR model kernel scale optimization possibility depends on the kernel function. For no-isotropic kernel function, the number of the kernel scale *l* is usually equal to the number of inputs. For isotropic kernel functions, the kernel scale *l* is selected Table 7) from a range of values calculated according to the following equation:(4)l=[min; max]=[0.001(max(X)−min(X));(max(X)−min(X))]
where max(X)—a maximum value from the input variable matrix, min(X)—a minimum value from the input variable matrix.

Different accuracy measures have been calculated from the experiments: Mean Squared Error (MSE), RMSE, and Mean Absolute Error (MAE) [32].

MSE is a measure representing the average of the squared difference between the real and predicted values of the data set. RMSE is simply the square root of the MSE, the only difference being that MSE measures the variance of the residuals, while RMSE measures the standard deviation of the residuals.
(5)$$\mathrm{RMSE}=\sqrt{\mathrm{MSE}},\;\mathrm{where}\;\mathrm{MSE}=\frac{1}{n}\sum_{t=1}^{n}{\left|y_{t}-{\hat{y}}_{t}\right|}^{2}$$
where *n*—the number of time points, $y_{t}$—is the actual value at a given time period *t*, and ${\hat{y}}_{t}$—is the predicted value, *t*—observation in a dataset.

The value of RMSE and MSE penalizes large errors. In contrast, MAE is less biased for higher values and usually does not penalize large errors. MAE is calculated according to the following equation:(6)$$\mathrm{MAE}=\frac{1}{n}\overset{n}{\underset{t=1}{\sum }}\left|y_{t}-{\hat{y}}_{t}\right|$$
where *n*—the number of time points, $y_{t}$—is the actual value at a given time period *t*, and ${\hat{y}}_{t}$—is the predicted value.

Figure 8 shows the minimum MSE of the GPR algorithm, where the red dot indicates the iteration with the minimum MSE, and the light blue dot represents the computed MSE value during the optimization process by varying the GPR hyperparameters. Dark blue dots indicate the observed minimum error minMSE detected up to the current (including current as well) observation:(7)$$\mathrm{minMSE}=\min\left({\mathrm{MSE}}^{i}\right),i=\bar{1,n},\;$$
where *n* is the number of iterations.

Figure 9 shows the cross-validation results depicting the predicted value of VFC vibro excitation against the real (true) values. The errors are represented by vertical red dashed lines, but due to very small error values, the majority of the true and predicted value points overlap.

The best results achieving RMSE=5.6891, MSE=31.238, and MAE=3.872 have been achieved using a linear basic function, no-isotropic rational quadratic kernel function (Equation (8)) with sigma 0.0002.
(8)$$k\left(x_{a},x_{b}\right)=\sigma^{2}{\left(1+\frac{{\left||x_{a}-x_{b}|\right|}^{2}}{2\alpha \ell^{2}}\right)}^{-\alpha }$$
where $\sigma^{2}$—the overall variance, ℓ—the length scale parameter, α—the scale-mixture (α>0).

Four hyperparameters have been included in the SVM model optimization process: kernel function, kernel scale, box constraint, and epsilon (Table 8). The ranges of values for the latter three hyperparameters were selected on the basis of preliminary experiments. Seven different kernel functions have been analyzed: three Gaussian (fine, medium, coarse), Linear, Quadratic, and Cubic. It has been observed that the Gaussian functions gave the poorest results compared to other functions.

The best results for the validation set—RMSE = 9.124, MSE = 83.253, and MAE = 6.403—were obtained using a linear kernel function, ε = 0.105, with box constrains = 111.25 (Figure 10).

It can be noted that the minimum MSE varies over a wide range depending on the combination of the SVM hyperparameters, and the error can reach almost 3000. Prediction errors are displayed in a response plot in Figure 11.

For the ANN model experimental setup, we have used a simple feedforward network—multilayer perceptron (MLP), presented in Figure 12.

As in the two previous ANN models, four hyperparameters were used in the optimization process: the number of hidden layers, the size of the hidden layer, the activation function of the hidden layers, and regularization strength (Table 9). The three most common activation functions were analyzed: Sigmoid, Hyperbolic tangent (Tanh), and Rectified Linear Unit (ReLU). The range of regularization strength was chosen based on primary cross-validation results. Value ranges of hidden number layers and hidden layer size were selected according to the size of the data set.

The ANN approach provides very good prediction accuracy and the best results with RMSE = 4.5337, MSE = 20.573, and MAE = 3.528 were obtained using a single hidden layer neural network with ReLU activation function, 12 neurons, and a regularization strength of zero. The variation of the minimum MSE values during the ANN hypermeter optimization process is shown in Figure 13. The testing data results of the ANN model are presented in Figure 14 providing actual and predicted values of VFC vibration excitation.

Thus, the minimum leaf size is the only hyperparameter that was included in the optimization. It denotes the minimum number of data points that are required to be present in the leaf node. The search range for this hyperparameter is from 1 to 15 which is chosen according to the size of the data set. The best result of the DT approach: RMSE = 29.567, MSE = 874.19, and MAE = 21.507 (Figure 15) were obtained using a decision tree with a minimum leaf size equal to four. The decision tree approach provides the lowest accuracy compared to GPR, SVM, and ANN. The minimum MSE value varies from 2195.31 to 874.19. As depicted in Figure 16, in half of the observations, the distance between the predicted VFC vibration excitation value and the actual values is more significant than GPR, ANN, or SVM.

Table 10 represents the hyperparameters of another, i.e., the KNN approach. Two KNN hyperparameters were included in the optimization process. The first one is K (the number of neighbors to consider) and the second is the employed distance function (most commonly used Euclidean, Manhattan, Minkowski).

For *n*-dimensional space, the Euclidean distance between the two points x with coordinates $\left(x_{1},x_{2},\ldots ,x_{n}\right)$ and y with coordinates $\left(y_{1},y_{2},\ldots ,y_{n}\right)$ is determined using the following equation:(9)$$d_{Eucl}\left(x,y\right)=\sqrt{{\left(x_{1}-y_{1}\right)}^{2}+{\left(x_{2}-y_{2}\right)}^{2}+\ldots +{\left(x_{n}-y_{n}\right)}^{2}}=\sqrt{\overset{n}{\underset{i=1}{\sum }}{\left(x_{i}-y_{i}\right)}^{2}}$$
where $\left(y_{1},y_{2},\ldots ,y_{n}\right)$ are attribute values of *y* data instance and $\left(x_{1},x_{2},\ldots ,x_{n}\right)$ are attribute values of *x* data instance.

The Manhattan distance is also known as city block distance, or taxicab geometry, as well as several other names, because it allows calculating the distance between two data points on a uniform grid, for example, a city block; there may be more than one path between the two points that have the same Manhattan distance. The Manhattan distance between two points x and y is calculated using the formula:(10)$$d_{Manh}\left(x,y\right)=\overset{n}{\underset{i=1}{\sum }}\left|x_{i}-y_{i}\right|$$

Minkowski distance is a generalized distance metric. The above formula (Equation (10)) can be manipulated by substituting ‘*p*’ to calculate the distance between two data points in different ways. Thus, Minkowski distance is also known as *Lp* norm distance:(11)$$d_{Mink}\left(x,y\right)={\left(\overset{n}{\underset{i=1}{\sum }}{\left|x_{i}-y_{i}\right|}^{p}\right)}^{1/p}$$
where p is the order of the Minkowski metric. With different values of p, the distance between two data points can be calculated in different ways: p=1—Manhattan distance; p=2—Euclidean distance, p=∞—Chebyshev’s distance. A value such as p=1.5 provides a balance between the two measures.

The best KNN results, including RMSE = 6.0757, MSE = 36.915, and MAE = 3.528 were obtained using the Manhattan distance for two neighbors, *K* = 2 (Figure 17).

The most important step in KNN is to determine the optimal value of *K*. The optimal value of *K* reduces the effect of noise on the classification. A technique called the “elbow method” helps to do this, selecting the optimal *K* value. Different values of *K* are applied to the same data set and the change in *K* is initially observed. In the data set characterizing the SPIF process, the error rate (RMSE) curve obtained by applying the KNN with respect to the *K* value is shown in Figure 18.

The graph presented in Figure 18 denotes that initially the error rate decreases to 2, and then it starts to increase. Thus, the value of *K* should be 2, i.e., it is the optimal *K* value for this model. This curve is called an elbow curve because it has a shape of an elbow and is commonly used to adjust the *K* value.

*R*^2^ (coefficient of determination) is a regression score, which is a statistical measure indicating how close the data are to the fitted regression line. In regression, it is a measure showing how well the regression predictions approximate the real data. An *R*^2^ of 1 indicates that the regression predictions perfectly fit the data:(12)$$R^{2}=\frac{SSR}{SST}=\frac{\sum_{i=1}^{m}{\left(y_{i}-\hat{y_{i}}\right)}^{2}}{\sum_{i=1}^{m}{\left(y_{i}-\bar{y}\right)}^{2}}$$
where *SSR* is the sum of squares of residuals, *SST* is the total sum of squares, $y_{i}$ is the actual value, $\hat{y_{i}}$ is the predicted value, and $\bar{y}$ is the mean value.

*R*^2^ is always between 0 and 100% (or 0 and 1.0). The higher the *R^2^*, the better the model. The goal is not to maximize *R*^2^ because model stability and adaptability are equally important. When checking the adjusted *R*^2^ value, it is preferred to have the values of the *R*^2^ and adjusted *R*^2^ close to each other. From the graphical representation of *R*^2^ values of the five prediction models (Figure 19), it can be seen that the DT algorithm gave the worst result (*R*^2^ = 0.878) compared to others.

A series of five experimental runs have been carried. Summarizing the experimental results, ANN and GPR were identified as the most efficient methods for developing VFC vibro excitation prediction models, giving the lowest prediction error (RMSE) of 4.5337 and 5.6891, respectively (Figure 20). It should be noted that the DT algorithm is inappropriate for this task and for the available data set, as the prediction errors in both cases (with and without optimization) are high, reaching around 30%. As for the standard deviation (ST), despite the DT model with a 0 value of ST for all five iterations, the GDR has the lowest standard deviation, ST = 0.201. The ANN model has resulted with ST = 0.616, KNN with ST = 0.78, and the SVM with the highest value, ST = 2.531.

## 5. Discussion

The innovative method and technique of the SPIF process proposed in this paper have made it possible to dispense with the environmentally unfriendly process of lubrication of the metal sheet surface and replace it by ultrasonic excitation in the plane of the sheet in two perpendicular directions. In order to determine the most effective frequency range for ultrasonic excitation of metal sheets, a 3D scanning experiment was carried out (Figure 1). The results of the experiment (Figure 2) show that the SPIF process is the most efficient in the frequency range of 28–36 kHz of ultrasonic vibrations, where the vibrations in the XY sheet plane directions dominate vis-a-vis normal vibrations in the Z direction. An advanced surface roughness tester (TR200 Time group) was used to measure the roughness of the formed sheet surface. The surface roughness of the sheet formed without ultrasonic vibration was measured to be Ra = 0.30–0.33 μm and Ra = 0.18–0.25 μm with ultrasonic vibration. The surface roughness results obtained were compared with the SPIF results of lubricated surfaces reported in [1], which showed that lubricants have poor oxidative stability, which leads to changes in the physical properties of the lubricant, such as viscosity, acidity, etc. Therefore, the change in these physical properties and the detachment of the alumina from the AA1100 aluminum alloy formed samples may lead to an increase in the force and friction coefficient values, and the surface roughness, which was set to be close to Ra = 1.45 μm. The paper [22] confirms that the coefficient of friction between ultrasonically excited metal contact surfaces is reduced by up to 80%. Experimental results carried out employing data exploration techniques show that two features, namely ‘Tool sphere diameter’ and ‘Wall angle’, have constant values and need to be eliminated. It was also found that the correlation coefficient between ‘Tool type’ and the output ‘VFC vibro excitation’ is not very significant (*r* = −0.2694) and is the lowest compared to the other features, and can be removed as well. Various ML-based algorithms including GPR, SVM, DT, KNN, and ANN have been used for the prediction task, performing Bayesian hyperparameter optimization. Summarizing the experimental results, it is found that ANN and GPR are the most efficient methods for developing VFC vibro excitation prediction models, with the lowest prediction error (RMSE) of 4.5337 and 5.6891, respectively. However, comparing all three metrics—RMSE value, the standard error deviation, and execution time, the GPR shows superiority over the ANN.

Different input functions and different test conditions make it difficult to compare experimental results with those obtained in different studies. However, evaluating the impact of different features, it can be concluded that the ‘Forming depth’ is one of the most important features in the force prediction process. Such findings are also observed in studies by other authors [15,21]. Nevertheless, in most cases, additional features with those correlation coefficients with the output features that are higher than *r* = ±0.3 allow for achieving better prediction accuracy. For comparison purposes, additional experiments were performed using all five input features and only three features, ignoring ‘Step depth’ and ‘Sheet thickness’ (Figure 21).

Comparing the RMSE value, it can be observed that the average prediction error for DT remains the same, with RMSE = 29.562. The KNN and SVM models based on input three provide higher RMSE values, with an increase of 1.939% and 13.67%, respectively, compared to the five-input. The most significant impact of denoted features elimination has been observed using ANN and GPR models because the RMSE of the three-input-based model increases more than 25% in both cases. This indicates that it is reasonable to estimate additional parameters in order to improve the accuracy of the prediction model.

## 6. Conclusions

An innovative way is proposed to reduce the frictional force between the forming tool and the sheet surface by exciting the sheet metal with ultrasonic vibrations in two rectangular directions of the sheet plane. The coefficient of friction between the ultrasonically excited metal sheet surface and the tool was found to be equal to the coefficient of friction between the lubricated surfaces, which speeds up the time to market and makes the process more environmentally friendly. Based on the experimentally determined values of the vertical forming forces, machine learning algorithms were developed to predict these forces. It has been demonstrated that ML algorithms can be successfully trained for the prediction of VFC vibro excitation on a relatively small data set and provide good generalization performance on test samples. Five different ML algorithms (ANN, SVM, GPR, KNN, and DT) were used in the experimental research, including the process of hyperparameters’ optimization. The obtained results verified the practical observations of other authors working on similar problems, confirming that ANN is the most appropriate algorithm, as it gives the lowest error with and without hyperparameters’ optimization: RMSE = 4.53 and RMSE = 5.87, respectively. In both experiments, GPR lags slightly behind ANN. However, the KNN algorithm after optimization showed a smaller prediction error (RMSE = 6.07) than SVM (RMSE = 9.12). Summarizing the accuracy results, ANN and GPR have been identified as the most efficient methods for developing VFC vibro excitation prediction models, although their training time differs significantly. In his case, the training time is less important than the prediction accuracy results, while ANN has a 20% accuracy advantage, which is a more important factor.

## Figures and Tables

**Figure 1 sensors-22-00018-f001:**
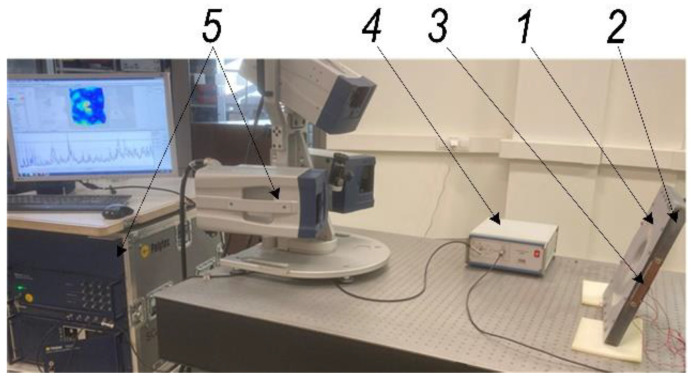
3D scanning experimental set-up for the investigation of vibration modes of the metal sheet rigidly attached to the frame: 1—an experimental body—aluminum alloy sheet; 2—steel base frame; 3—piezoelectric actuator; 4—liner amplifier P200 (FLC Electronics AB, Sweden); 5—3D scanning vibrometer PSV-500-3D-HV (Polytec GmbH, Germany).

**Figure 2 sensors-22-00018-f002:**
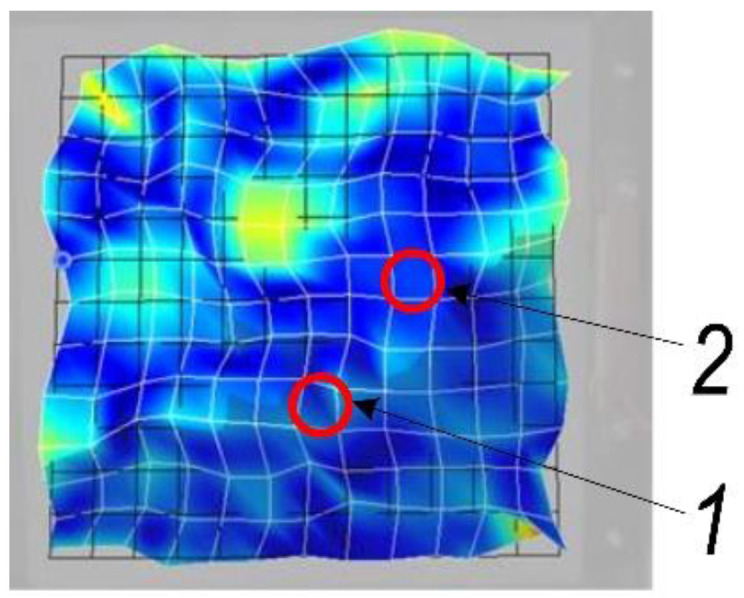
An aluminum alloy sheet vibrations at an excitation frequency of 30 kHz and forming tool contact with a sheet location: 1—bottom; 2—on the side.

**Figure 3 sensors-22-00018-f003:**
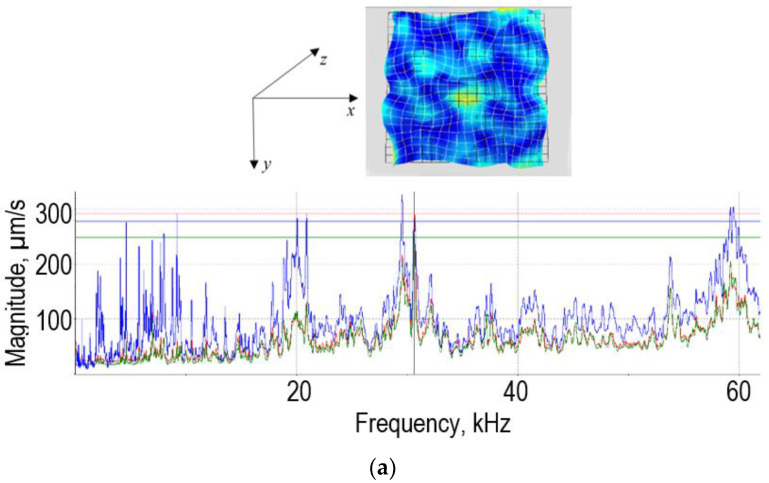
3D frequency response and deformations of an aluminum alloy sheet in the frequency range from 0.5 to 60 kHz measured with a Polytec PSV-500 3D laser Dopler vibrometer scanner: x (red) and y (green) are lateral vibrations, z (blue) in the normal direction, respectively: influence of tool position and forming depth on the eigenmodes of the sheet at tool position 1, forming depth: (**a**) tool position 1, forming depth 15 mm; (**b**) tool position 2, forming depth 30 mm.

**Figure 4 sensors-22-00018-f004:**
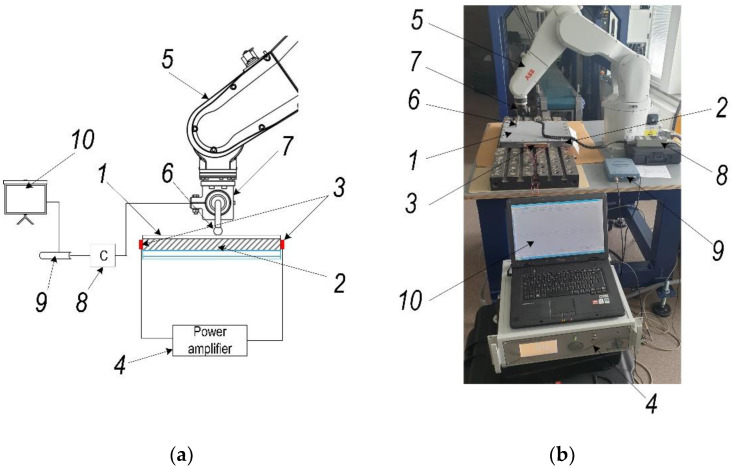
Experimental set-up for sample excitation: 1—aluminum alloy sheet; 2—metal frame; 3—piezoelectric transducers; 4—power amplifier; 5—robotic arm; 6—forming tool with sphere; 7—mechanical momentum sensor STJ100; 8—controller BGI (Mark-10 Corp., Copiague, NY, USA); 9—oscilloscope PicoScope 3424 (Pico Technology Ltd., St. Neots, UK, GB); 10—PC: (**a**) scheme; (**b**) experimental setup view.

**Figure 5 sensors-22-00018-f005:**
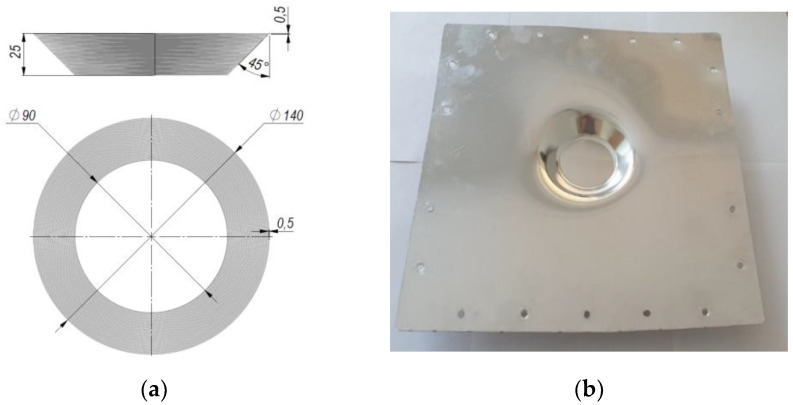
Incremental forming of aluminum sheet: (**a**) An outline of the used geometric parameters; (**b**) photo of the formed product.

**Figure 6 sensors-22-00018-f006:**
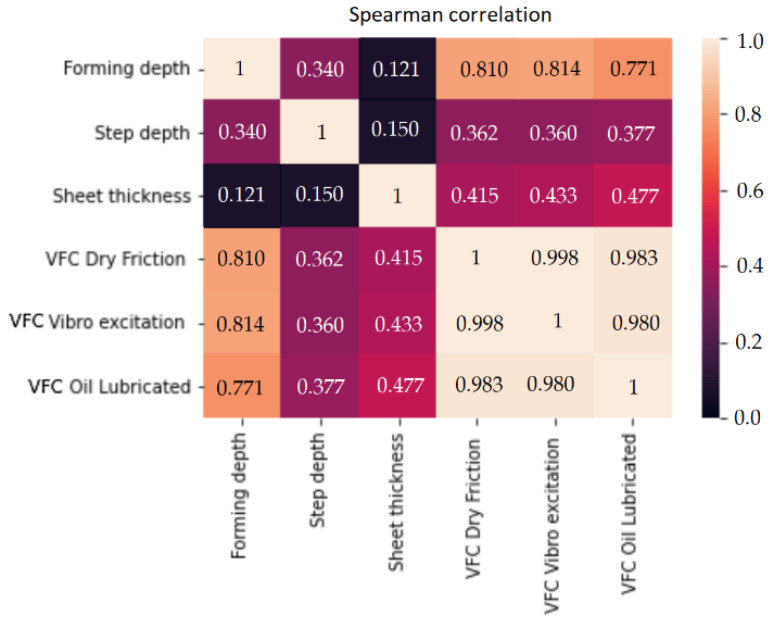
Spearman correlation matrix and *r* coefficient.

**Figure 7 sensors-22-00018-f007:**
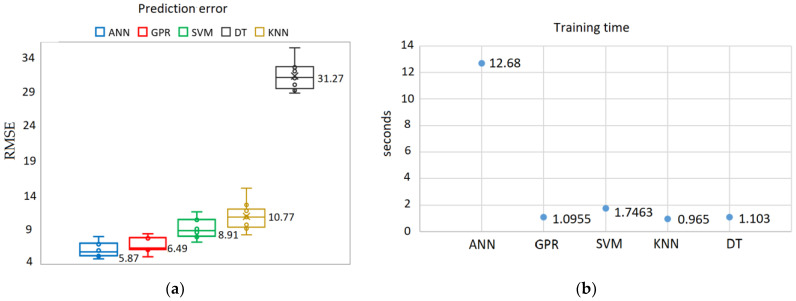
Prediction results using different machine learning algorithms: ANN, GPR, SVM, KNN, DT: (**a**) prediction error suing k-folds cross-validation, where k = 10; (**b**) training time results.

**Figure 8 sensors-22-00018-f008:**
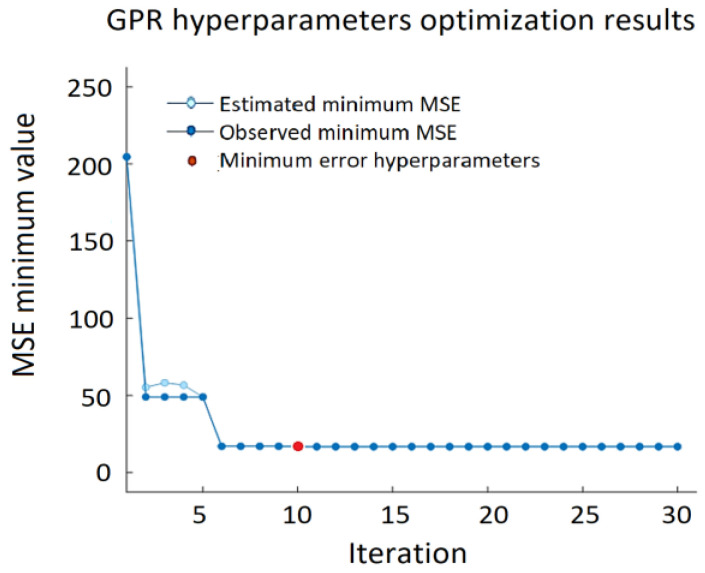
Minimum MSE during GPR hypermeters’ optimization process.

**Figure 9 sensors-22-00018-f009:**
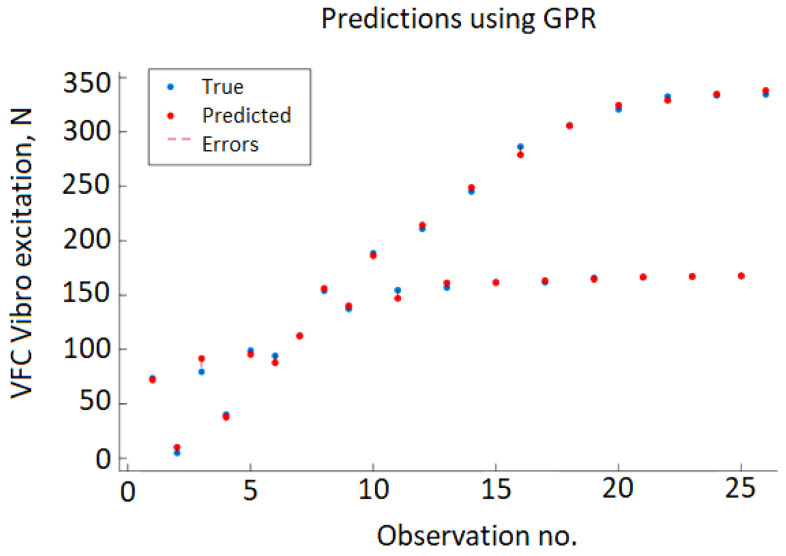
Testing data results of the GPR model.

**Figure 10 sensors-22-00018-f010:**
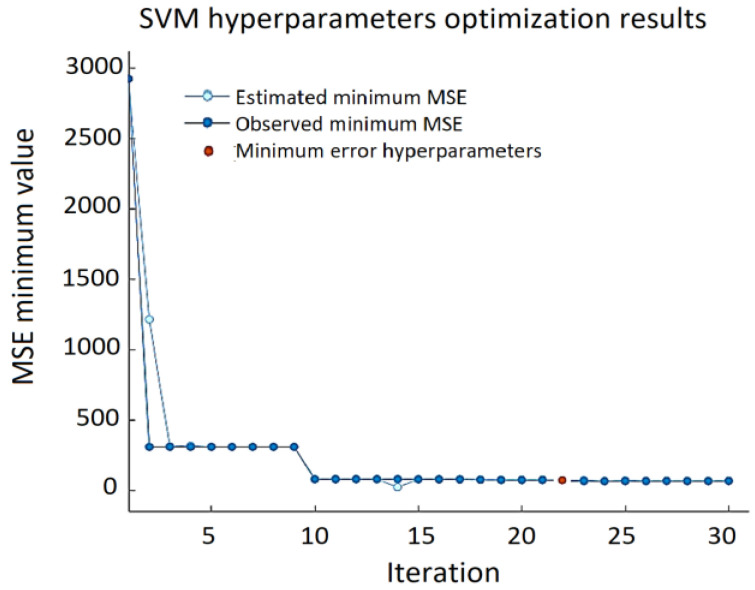
Minimum MSE during SVM hypermeters’ optimization process.

**Figure 11 sensors-22-00018-f011:**
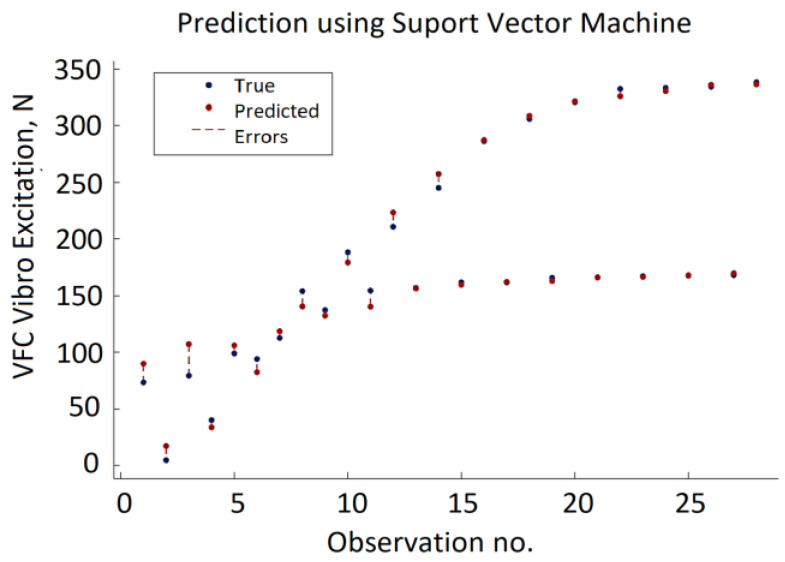
Testing data results of the SVM model.

**Figure 12 sensors-22-00018-f012:**
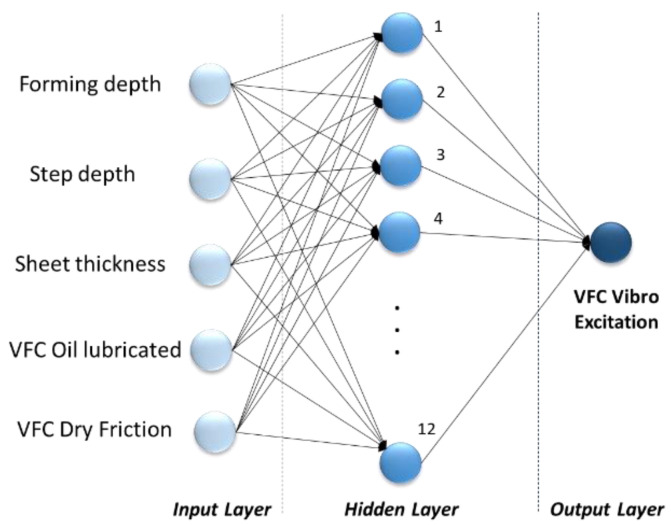
The architecture of used multilayer perceptron.

**Figure 13 sensors-22-00018-f013:**
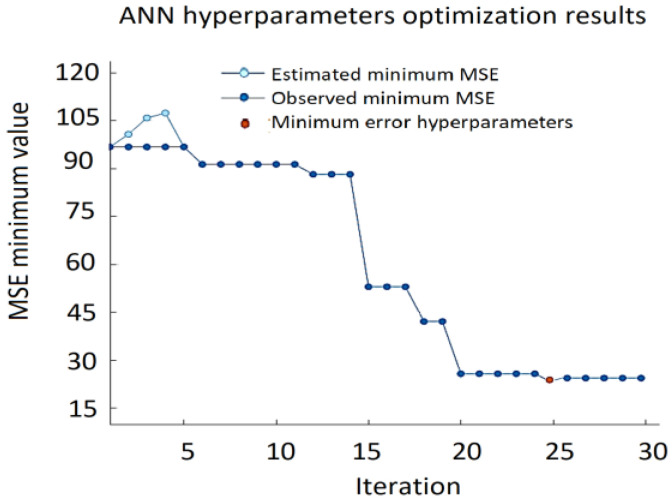
Minimum MSE during ANN hypermeters’ optimization process.

**Figure 14 sensors-22-00018-f014:**
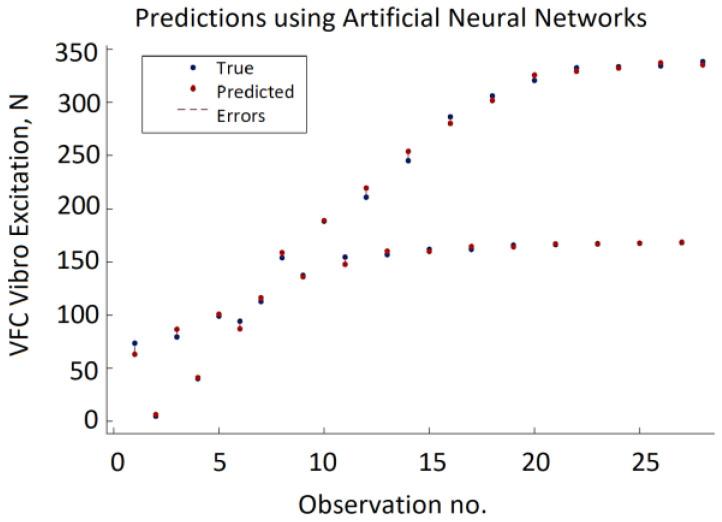
Testing data results of the ANN model.

**Figure 15 sensors-22-00018-f015:**
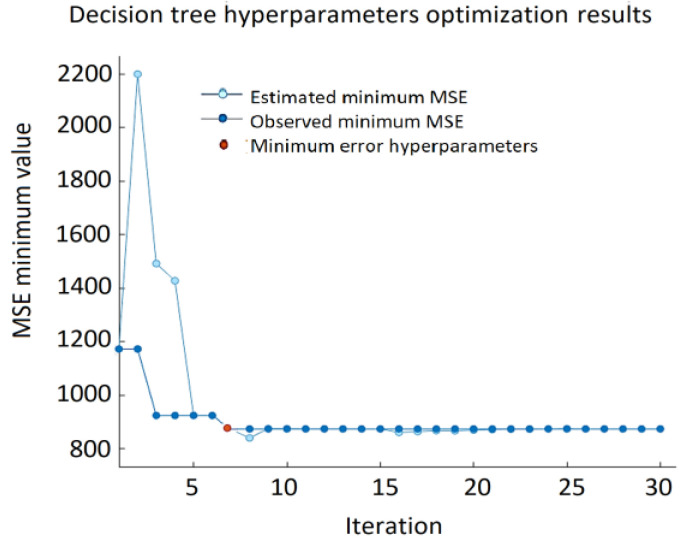
Minimum MSE during DT hypermeters’ optimization process.

**Figure 16 sensors-22-00018-f016:**
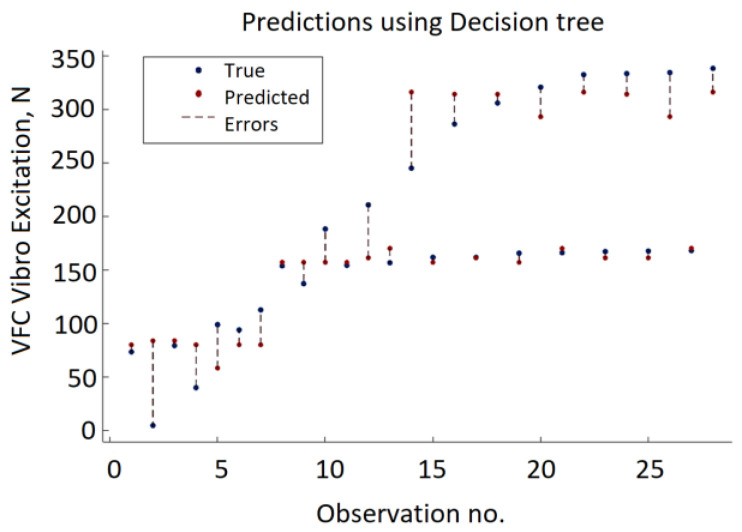
Testing data results of the DT model.

**Figure 17 sensors-22-00018-f017:**
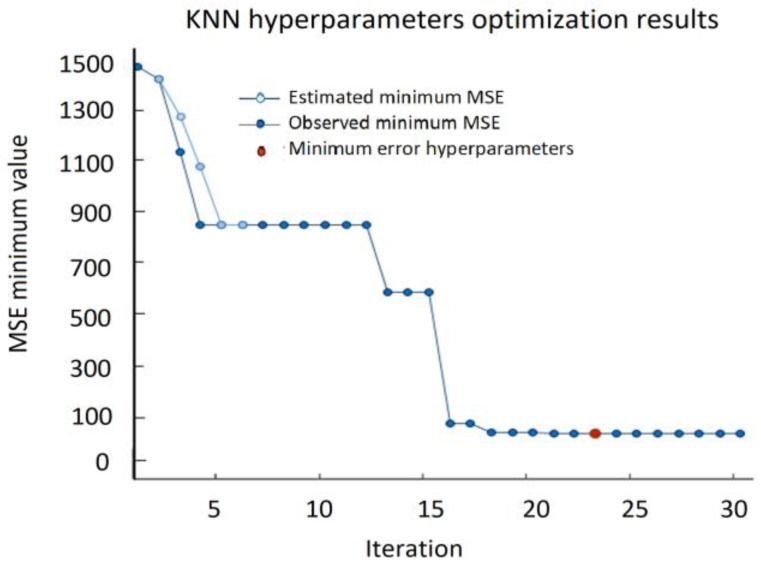
Minimum MSE during KNN hypermeters’ optimization process.

**Figure 18 sensors-22-00018-f018:**
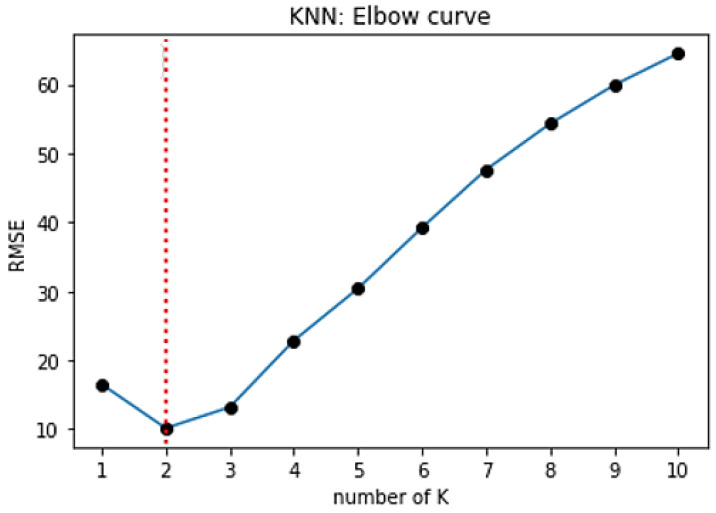
A visual curve with an explicit elbow point in the K range [1; 10].

**Figure 19 sensors-22-00018-f019:**
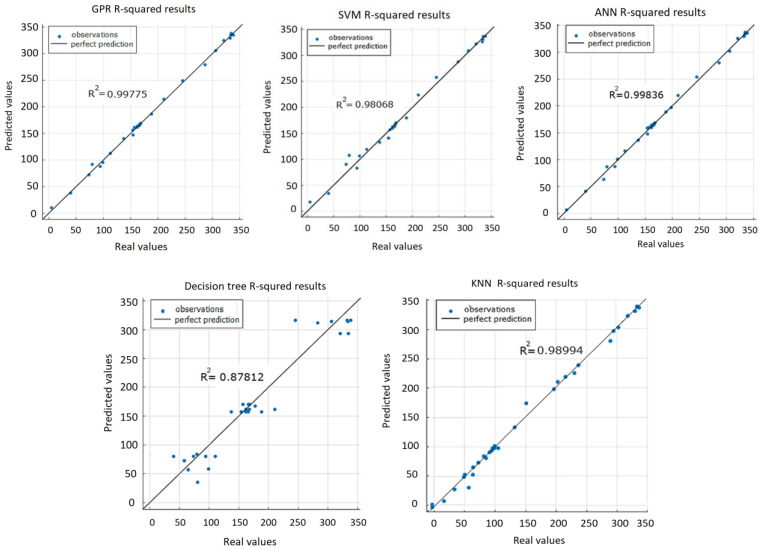
R-squared representation for all five ML algorithms used for the prediction task.

**Figure 20 sensors-22-00018-f020:**
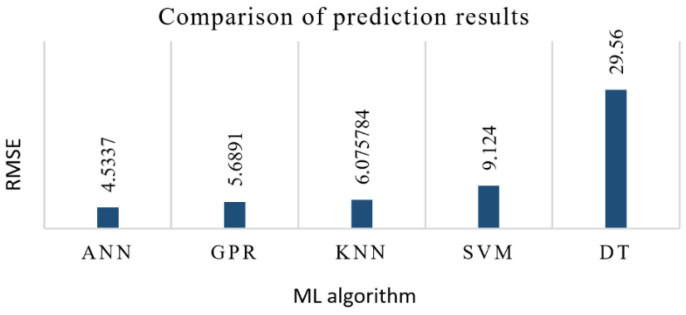
Comparison of different ML algorithm prediction error results.

**Figure 21 sensors-22-00018-f021:**
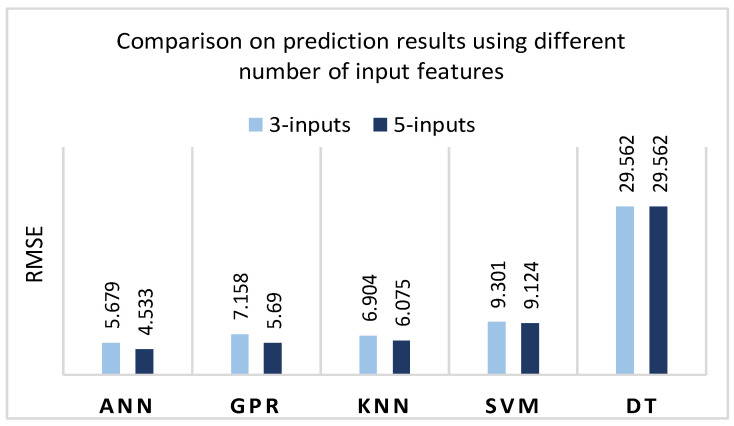
Comparison of prediction results using a different number of input features: five input features (forming depth, step depth, sheet thickness, VFC dry friction, VFC vibro excitation) and three inputs (forming depth, VFC dry friction, VFC vibro excitation).

**Table 1 sensors-22-00018-t001:** Mechanical properties and chemical composition of aluminum alloy AW1050, used in experiments.

Parameter	Value	Unit
Proof stress	85 min	MPa
Tensile strength	105–145	MPa
Hardness Brinell	34	HB
Elongation A	12 min	%
Density	2.71	kg/m^3^
Melting point	650	°C
Modulus of Elasticity	71	GPa
Electrical Resistivity	0.282 × 10^−6^	Ω·m
Thermal Conductivity	222	W/m·K
Thermal Expansion	24 × 10^−6^	/K
**Chemical composition**		
**Element**	**% Weight**	
Aluminum	99.5	
Silicon	0–0.25	
Iron	0–0.4	
Magnesium	0–0.05	
Manganese	0–0.05	
Titanium	0–0.03	
Vanadium	0–0.05	
Copper	0–0.05	

**Table 2 sensors-22-00018-t002:** Parameters of tool forming path (trajectory).

Parameter	Value	Unit
Type of trajectory	Helix—Circle	-
Helix type	Right-handed	-
Lead angle	45	°
Step size to the centre, used in the experiment	0.5	mm
Step size downwards, used in the experiment	0.5	mm
Major diameter of the helix	140	mm
Minor diameter of the helix	9	mm
Maximum possible depth	2	mm
Depth, used in the experiments	25	mm
Radius of the tool sphere	8.5	mm
Possible step size to the centre interval	0.1–1.0	mm
Possible step size downwards interval	0.1–1.0	mm

**Table 3 sensors-22-00018-t003:** Friction coefficient and friction force averaged measurement results.

Method	Friction Force	Friction Coefficient
Without lubrication and vibration	3.2 N	0.5
With lubrication	1.6 N	0.1
With vibration	1.9 N	0.12

**Table 4 sensors-22-00018-t004:** Parameters for data exploration.

No.	Input Parameter	Value (min, max)	Unit
1	Forming depth	0 … 26	mm
2	Tool end diameter	20 … 20	mm
3	Step depth (Δz)	0.25 … 0.5	mm
4	Step width (Δx)	0.25 … 0.5	mm
5	Wall angle	45 … 45	°
6	Sheet thickness (Θ, °)	0.5 … 0.8	mm
7	Tool type	rotating sphere on the end/ not rotating	-
8	Vertical Force Component (VFC) dry friction	4.91 … 343.35	N
9	VFC oil lubricated	1.96 … 341.39	N
**Output parameter**			
1	VFC vibro excitation	4.91…338.45	N

**Table 5 sensors-22-00018-t005:** Training parameters.

**Input Parameters**				
Forming depth	Step depth (Δz)	Step width (Δx)	Sheet thickness (Θ, °)	Tool type
**Output Parameters**				
VFC vibro excitation				

**Table 6 sensors-22-00018-t006:** Vertical force dependence from different friction when sheet thickness is 0.5 mm.

	s = 0.5 mm, Steps 0.5 and 0.5 mm, Velocity Varied to 100%		
Forming Depth, mm	Vertical Force, N		
	Dry Friction	Oil Lubricated	Vibro Excitation
0	98.10	80.93	73.58
2	98.10	115.76	79.46
4	112.82	100.55	99.08
6	116.74	117.72	112.82
8	138.32	127.04	137.34
10	144.21	141.07	154.51
12	159.41	146.17	156.96
14	166.28	148.33	161.87
16	166.08	149.60	161.87
18	167.75	151.86	165.79
20	169.22	152.35	166.28
22	169.71	152.84	167.26
24	171.18	153.23	167.75
25	173.15	153.33	168.24

**Table 7 sensors-22-00018-t007:** Hyperparameters of the GPR model.

	Sigma	Basic Function	Kernel Function	Kernel Scale
**Value Range**	(0.0001; 948.39)	Constant; Zero; Linear	Isotropic and No-isotropic Exponential; Isotropic and No-isotropic Matern 3/2 and 5/2; Isotropic and No-isotropic Rational Quadratic; Isotropic and No-isotropic Squared Exponential	(0.33943–339.43)

**Table 8 sensors-22-00018-t008:** Hyperparameters of the SVM model.

	Epsilon	Box Constraint	Kernel Function	Kernel Scale
**Value Range**	(0.01; 1000)	(0.001; 500)	Gaussian, Linear, Quadratic, and Cubic	(0.001; 100)

**Table 9 sensors-22-00018-t009:** Hyperparameters of the ANN model.

	Number of Hidden Layers	Hidden Layer Size	Activation Function	Regularization Strength
**Value Range**	(1; 3)	(10; 100)	Sigmoid, Tanh, ReLU	(0; 0.001)

**Table 10 sensors-22-00018-t010:** Hyperparameters of the KNN model.

	K	Distance Metric
**Value Range**	(1; 10)	Euclidean, Manhattan, Minkowski (*p* = 1, *p* = 1.5, *p* = 2, *p* = *Infinity*)

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
