# Peer review of "Comparative Analysis of Machine Learning Methods for Predicting Robotized Incremental Metal Sheet Forming Force"

_sensors, 2021, doi:10.3390/s22010018_

Round 1

Reviewer 1 Report

Review “Comparative analysis of machine learning methods for predicting robotized incremental metal sheet forming force”

The present paper provides information about a method for predicting forces by extracting data from the characteristics of the Single Point Incremental Forming (SPIF) process. A technique based on several machine learning (ML) algorithms were applied to estimate the process forces. Although the current study is very interesting, a few things need to be improved before publication.

  • At first, the introduction needs to be improved. It needs to contain some results.
  • The introduction section is too lengthy for such a small number of references. The authors need to add several research articles to the introduction section in order to enrich it.
  • In the last paragraph of the mentioned above section, the authors have to mention the importance of developing and applying this method.
  • In the materials section, the authors need to add tables with the mechanical and chemical properties of the aluminum alloy.
  • The font size of the axis titles and the details in the figures (in figures 3, 7b, 8, 9, 10, 11, 14, and 15) should be larger.
  • There are several abbreviations. In my humble opinion, the authors need to add a nomenclature.
  • In general, the paper needs to be prepared according to the new MDPI template. The current paper lacs of a discussion section where the results are compared with relevant studies.

In conclusion, the paper needs major revision.

Author Response

Dear reviewer, thank you for your time and quality evaluation of the publication. We are are sending answers to your questions.

Reviewer 2 Report

This paper presents an approach based on machine learning algorithms to predict the SPIF forces. The research has a certain application value for practical engineering. However, some issues need to be addressed. The comments are as follows:

  • In Section 2, it is not clear how does the 3D scanning device measured the vibration of the metal sheet. What’s the condition used for Fig. 2 and Fig. 3? What’s the size of the aluminum alloy sheet and the tool? The authors should describe the detail information.
  • The sentence “Ultrasonic assisted friction reduction is well known in the field ofmetal-to-metal contacts.” The author should add relevant references to further verify this.
  • The principle for the employment of vibrationally excited metal sheet is not described clearly. A schematicdiagram is recommended to better illustrate the principle.
  • Table 3 shows only one set of data for each situation. To avoid chance factor during measurement, the author should better provide multiple groups of repeated detection data.
  • In Figs. 8-15, all the true values seem to be nearly the same with the predicted values. What’s the testing numbers for each point, and What is the standard deviation.

Author Response

(The authors gave the same response as above.)

Reviewer 3 Report

This paper presents a comparative study of different machine learning methods for predicting the metal sheet forming force. The paper includes many technical details that can be easily followed by the audience. The biggest inadequacy is that, there is no obvious algorithm development as compared with the state-of-the-art. It seems that only the performances of different “standard” algorithms are put into comparison, upon which the best one is identified. Some comments are provided below to be addressed.

Abstract is not well-written, which doesn’t accurately summarize the work and highlight the novelty of the work. Try to include the keywords in the abstract if possible.

Page 1, “methodological” is not necessarily added before “methods”.

In introduction, before the last paragraph, it is necessary to summarize the challenge and limitation of current studies instead of just list all others’ works. This naturally draws the objective of this research.

More related references are supposed to be given in Introduction to show the comprehensive literature survey.

The label and legend font sizes in a lot of figures are too small to read. They need to be enlarged, such as Figure 4a, Figure 10, Figure 12 and so on. The suggestion is to keep the same and large label sizes for all figures presented.

Check the consistency of equations sizes, e.g., Equation (1) seems to have a different size.

What is the full name of “KNN”? it needs to be spelled out at first time showing up.

What is the specific type of ANN? ANN is a neural network from a broad perspective, which can be any neural network with any customized architecture, such as CNN? LSTM? MLP? Need to specify which neural network and its layer configuration are utilized.

Pearson coefficient is a metric to evaluate the linearity of two variables. There indeed exist other important nonlinear relation. Authors may comment why they use this metric in this particular case. For example, whey they didn’t use the Spearman coefficient? Also, check if Figure 6 is correct. Figure 6 represents the correlation matrix of outputs not the correlation matrix between inputs and outputs. Thus, how r=0.2694 that indicates the relation between tool type and VFC… comes from?

In Section 4.1, authors said that the hyperparameter optimization is implemented through Bayesian optimization. The related information needs to be given. In addition, the hyperparameters of different models should be listed. It seems only the hyperparameters of GPR are given. Is the noise term added in the covariance kernel of GPR?

Author Response

(The authors gave the same response as above.)

Round 2

Reviewer 1 Report

At first, I would like to thank the authors for improving their manuscript according to most of my comments. However, I have to point out that a couple of things still need to be improved to publish this manuscript.

There are small legends in the figures needing a larger font size, for example, figures 3, 7b, 8, 9, 10, 11, 14, 15, 16, and 19.

The discussion section needs to focus on explaining and evaluating what you found, showing how it relates to your literature review. In other words, the authors need to compare the published results by your colleagues with yours.

Author Response

Dear reviewer, thank you for your time and quality evaluation of the publication. We are answering your questions.

Reviewer 2 Report

The authors have revised the paper. Now it can be recommended for publication.

Author Response

Dear reviewer, thank you for your time and quality evaluation of the publication. 

Reviewer 3 Report

Authors' revisions have addressed my concerns. I agree to accpet the paper.

Author Response

(The authors gave the same response as above.)
